# LENGTH-ADAPTIVE TRANSFORMER: TRAIN ONCE WITH LENGTH DROP, USE ANYTIME WITH SEARCH

## ABSTRACT

Although transformers have achieved impressive accuracies in various tasks in natural language processing, they often come with a prohibitive computational cost, that prevents their use in scenarios with limited computational resources for inference. This need for computational efficiency in inference has been addressed by for instance PoWER-BERT (Goyal et al., 2020) which gradually decreases the length of a sequence as it is passed through layers. These approaches however often assume that the target computational complexity is known in advance at the time of training. This implies that a separate model must be trained for each inference scenario with its distinct computational budget. In this paper, we extend PoWER-BERT to address this issue of inefficiency and redundancy. The proposed extension enables us to train a large-scale transformer, called *Length-Adaptive Transformer*, once and uses it for various inference scenarios without re-training it. To do so, we train a transformer with *LengthDrop*, a structural variant of dropout, which stochastically determines the length of a sequence at each layer. We then use a multi-objective evolutionary search to find a length configuration that maximizes the accuracy and minimizes the computational complexity under any given computational budget. Additionally, we significantly extend the applicability of PoWER-BERT beyond sequence-level classification into token-level classification such as span-based question-answering, by introducing the idea of *Drop-and-Restore*. With Drop-and-Restore, word-vectors are dropped temporarily in intermediate layers and restored at the last layer if necessary. We empirically verify the utility of the proposed approach by demonstrating the superior accuracy-efficiency trade-off under various setups, including SQuAD 1.1, MNLI-m, and SST-2. Upon publication, the code to reproduce our work will be open-sourced.

## 1 INTRODUCTION

Pretrained language models (Peters et al., 2018; Devlin et al., 2018; Radford et al., 2019; Yang et al., 2019) have achieved notable improvements in various natural language processing (NLP) tasks. Most of them rely on transformers (Vaswani et al., 2017), and the number of model parameters ranges from hundreds of millions to billions (Shoeybi et al., 2019; Raffel et al., 2019; Kaplan et al., 2020; Brown et al., 2020). Despite this high accuracy, excessive computational overhead during inference, both in terms of time and memory, has hindered its use in real applications. This level of excessive computation has further raised the concern over energy consumption as well (Schwartz et al., 2019; Strubell et al., 2019).

Recent studies have attempted at addressing these concerns regarding large-scale transformers' computational and energy efficiency (see §6 for a more extensive discussion.) Among these, we focus on PoWER-BERT (Goyal et al., 2020) which progressively reduces sequence length by eliminating word-vectors based on the attention values as passing layers. PoWER-BERT establishes the superiority of accuracy-time trade-off over earlier approaches (Sanh et al., 2019; Sun et al., 2019; Michel et al., 2019). It however requires us to train a separate model for each efficiency constraint. In this paper, we thus develop a framework based on PoWER-BERT such that we can train a single model that can be adapted in the inference time to meet any given efficiency target.

In order to train a transformer to cope with a diverse set of computational budgets in the inference time, we propose to train one while reducing the sequence length with a random proportion at each

layer. We refer to this procedure as LengthDrop which was motivated by the nested dropout (Rippel et al., 2014). We can extract sub-models of shared weights with any length configuration without requiring extra post-processing nor additional finetuning.

Once a transformer is trained with the proposed LengthDrop, we search for the length configuration that maximizes the accuracy given a computational budget. Because this search is combinatorial and has multiple objectives (accuracy and efficiency), we use an evolutionary search algorithm, which further allows us to obtain a full Pareto frontier of accuracy-efficiency trade-off of each model.

It is not trivial to find an optimal length configuration given the inference-time computational budget, although it is extremely important in order to deploy these large-scale transformers in practice. In this work, we propose to use evolutionary search to find a length configuration that maximizes the accuracy within a given computational budget. We can further compute the Pareto frontier of accuracy-efficiency trade-off to obtain a sequence of length configurations with varying efficiency profiles.

PoWER-BERT, which forms the foundation of the proposed two-stage procedure, is only applicable to sequence-level classification, because by design it eliminates some of the word vectors at each layer. In other words, it cannot be used for token-level tasks such as span-based question answering (Rajpurkar et al., 2016), because these tasks require hidden representations of the entire input sequence at the final layer. We thus propose to extend PoWER-BERT with a novel Drop-and-Restore process (§3.3), which eliminates this inherent limitation. Word vectors are dropped and set aside, rather than eliminated, in intermediate layers to maintain the saving of computational cost, as was with the original PoWER-BERT. These set-aside vectors are then restored at the final hidden layer and provided as an input to a subsequent task-specific layer, which is unlike the original PoWER-BERT.

The main contributions of this work are two-fold. First, we introduce LengthDrop, a structured variant of dropout for training a single Length-Adaptive Transformer model that allows us to automatically derive multiple sub-models with different length configurations in the inference time using evolutionary search, without requiring any re-training. Second, we design Drop-and-Restore process that makes PoWER-BERT applicable beyond classification, which enables PoWER-BERT to be applicable to a wider range of NLP tasks such as span-based question answering. We empirically verify Length-Adaptive Transformer works quite well using the variants of BERT on a diverse set of NLP tasks, including SQuAD 1.1 (Rajpurkar et al., 2016) and two sequence-level classification tasks in GLUE benchmark (Wang et al., 2018). Our experiments reveal that the proposed approach grants us a fine-grained control of computational efficiency and a superior accuracy-efficiency trade-off in the inference time, compared to existing approaches.

## 2 BACKGROUND: TRANSFORMERS AND POWER-BERT

Before we describe our main approach, we review some of the building blocks in this section. In particular, we review transformers, which are a standard backbone used in natural language processing these days, and PoWER-BERT, which was recently proposed as an effective way to train a large-scale, but highly efficient transformer for sequence-level classification.

### 2.1 TRANSFORMERS AND BERT

A transformer is a particular neural network that has been designed to work with a variable-length sequence input and is implemented as a stack of self-attention and fully-connected layers (Vaswani et al., 2017). Here, we give a brief overview of the transformer which is the basic building block of the proposed approach.

Each token $x_t$ in a sequence of tokens $x = (x_1, \ldots, x_N)$, representing input text, is first turned into a continuous vector $h_t^0 \in \mathbb{R}^H$ which is the sum of the token and position embedding vectors. This sequence is fed into the first transformer layer which returns another sequence of the same length $h^1 \in \mathbb{R}^{N \times H}$. We repeat this procedure $L$ times, for a transformer with $L$ layers, to obtain $h^L = (h_1^L, \ldots, h_N^L)$. We refer to each vector in the hidden sequence at each layer as a *word vector* to emphasize that there exists a correspondence between each such vector and one of the input words.

Although the transformer was first introduced for the problem of machine translation, Devlin et al. (2018) demonstrated that the transformer can be trained and used as a masked language model.

More specifically, Devlin et al. (2018) showed that the transformer-based masked language model, called BERT, learns a universally useful parameter set that can be finetuned for any downstream task including sequence-level and token-level classification.

In the case of sequence-level classification, a softmax classifier is attached to the word vector $h_1^L$ associated with the special token `[CLS]`, and the entire network, including the softmax classifier and BERT, is finetuned. For token-level classification, we use each $h_t^L$ as the final hidden reprsentation of the associated $t$-th word in the input sequence. This strategy of pretraining followed by finetuning, often referred to as transfer learning, has recently become a dominant approach to classification in natural language processing.

## 2.2 PoWER-BERT

PoWER-BERT keeps only the topmost $l_j$ word vectors at each layer $j$ by eliminating redundant ones based on the significance score, which is the total amount of attention imposed by a word on the other words (Goyal et al., 2020). $l_j$ is the hyper-parameter determines how many vectors to keep at layer $j$. PoWER-BERT has the same model parameters with BERT, but the extraction layers are interspersed after the self-attention layer in every transformer blocks (Vaswani et al., 2017).

PoWER-BERT reduces inference time successfully, achieving better accuracy-time trade-off than DistilBERT (Sanh et al., 2019), BERT-PKD (Sun et al., 2019), and Head-Prune (Michel et al., 2019). Despite the original intention of maximizing the inference efficiency with the minimal loss in accuracy, it is possible to set up PoWER-BERT to be both more efficient and more accurate compared to the original BERT, which was observed but largely overlooked by Goyal et al. (2020).

Training a PoWER-BERT model consists of three steps: (1) finetuning, (2) length configuration search, and (3) re-training. The finetuning step is just like the standard finetuning step of BERT given a target task. A length configuration is a sequence of retention parameters $(l_1, \cdots l_L)$, each of which corresponds to the number of word vectors that are kept at each layer. These retention parameters are learned along with all the other parameters to minimize the original task loss together with an extra term that approximately measures the number of retained word vectors across layers. In the re-training step, PoWER-BERT is finetuned with the length configuration fixed to its learned one.

For each computational budget, we must train a separate model going through all three steps described above. Moreover, the length configuration search step above is only approximate, as it relies on relaxation of retention parameters which are inherently discrete. This leads to the lack of guaranteed correlation between the success of this stage and true run-time. Even worse, it is a delicate act to tune the length configuration given a target computational budget, because trade-off is *implicitly* made via a regularization coefficient. Furthermore, PoWER-BERT has an inherent limitation in that it only applies to sequence-level classification because it eliminates word vectors in intermediate layers.

## 3 LENGTH-ADAPTIVE TRANSFORMER

In this section, we explain our proposed framework which results in a transformer that reduces the length of a sequence at each layer with an arbitrary rate. We call such a resulting transformer a Length-Adaptive Transformer. We train Length-Adaptive Transformer with LengthDrop which randomly samples the number of hidden vectors to be dropped at each layer with the goal of making the final model robust to such drop in the inference time. Once the model is trained, we search for the optimal trade-off between accuracy and efficiency using multi-objective evolutionary search, which allows us to use the model for any given computational budget without finetuning nor re-training. At the end of this section, we describe Drop-and-Restore process as a way to greatly increase the applicability of PoWER-BERT which forms a building block of the proposed framework.

In short, we train a Length-Adaptive Transformer once with LengthDrop and Drop-and-Restore, and use it with an automatically determined length configuration for inference with any target computational budget, on both sequence-level and token-level tasks.

### 3.1 LENGTHDROP

Earlier approaches to efficient inference with transformers have focused on a scenario where the target computational budget for inference is known in advance Sanh et al. (2019); Goyal et al. (2020). This greatly increases the cost of deploying transformers, as it requires us to train a separate transformer for each scenario. Instead, we propose to train one model that could be used for a diverse set of target computational budgets without retraining.

LengthDrop randomly generates a length configuration by sequentially sampling a sequence length $l_{i+1}$ at the $(i+1)$-th layer based on the previous layer's sequence length $l_i$, following the uniform distribution $\mathcal{U}(\lceil (1-p)l_i \rceil, l_i)$, where $l_0$ is set to the length of the input sequence, and $p$ is the LengthDrop probability. This sequential sampling results in a length configuration $(l_1, \cdots, l_L)$. Length-Adaptive Transformer can be thought of as consisting of a full model and many sub-models corresponding to different length configuration, similarly to a neural network trained with dropout (Srivastava et al., 2014).

**LayerDrop**    From the perspective of each word vector, the proposed LengthDrop could be thought of as skipping the layers between when it was set aside and the final layer where it was restored. The word vector however does not have any information based on which it can determine whether it would be dropped at any particular layer. In our preliminary experiments, we found that this greatly hinders optimization. We address this issue by using LayerDrop (Fan et al., 2019) which skips each layer of a transformer uniformly at random. The LayerDrop encourages each word vector to be agnostic to skipping any number of layers between when it is dropped and when it is restored, just like dropout (Srivastava et al., 2014) prevents hidden neurons from co-adapting with each other by randomly dropping them.

**Sandwich Rule and Inplace Distillation**    We observed that standard supervised training with LengthDrop does not work well in the preliminary experiments. We instead borrow a pair of training techniques developed by Yu & Huang (2019) which are sandwich rule and inplace distillation, for better optimization as well as final generalization. At each update, we update the full model without LengthDrop as usual to minimize the supervised loss function. We simultaneously update $n_s$ randomly-sampled sub-models (which are called sandwiches) and the smallest-possible sub-model, which corresponds to keeping only $\lceil (1-p)l_i \rceil$ word vectors at each layer $i$, using knowledge distillation (Hinton et al., 2015) from the full model. Here, sub-models mean models with length reduction. They are trained to their prediction close to the full model's prediction (inplace distillation).

### 3.2 EVOLUTIONARY SEARCH OF LENGTH CONFIGURATIONS

After training a Length-Adaptive Transformer with LengthDrop, we search for appropriate length configurations for possible target computational budgets that will be given at inference time. The length configuration determines the model performance in terms of both accuracy and efficiency. In order to search for the optimal length configuration, we propose to use evolutionary search, similarly to Cai et al. (2019) and Wang et al. (2020). This procedure is efficient, as it only requires a single pass through the relatively small validation set for each length configuration, unlike re-training for a new computational budget which requires multiple passes through a significantly larger training set for each budget.

We initialize the population with constant-ratio configurations. Each configuration is created by $l_{i+1} = \lceil (1-r)l_i \rceil$ for each layer $i$ with $r$ so that the amount of computation within the initial population is uniformly distributed between those of the smallest and full models. At each iteration, we evolve the population to consist only of configurations lie on a newly updated efficiency-accuracy Pareto frontier by mutation and cross-over. Mutation alters an original length configuration $(l_1, \cdots, l_L)$ to $(l'_1, \cdots, l'_L)$ by sampling $l'_i$ from the uniform distribution $\mathcal{U}(l'_{i-1}, l_{i+1})$ with the probability $p_m$ or keeping the original length $l'_i = l_i$, sweeping the layers from $i = 1$ to $i = L$. Crossover takes two length configurations and averages the lengths at each layer. Both of these operations are performed while ensuring the monotonicity of the lengths over the layers. We repeat this iteration $G$ times, while maintaining $n_m$ mutated configurations and $n_c$ crossover'd configurations. Repeating this procedure pushes the Pareto frontier further to identify the best trade-off between two objectives, efficiency and accuracy, without requiring any continuous relaxation of length configurations nor using a proxy objective function.

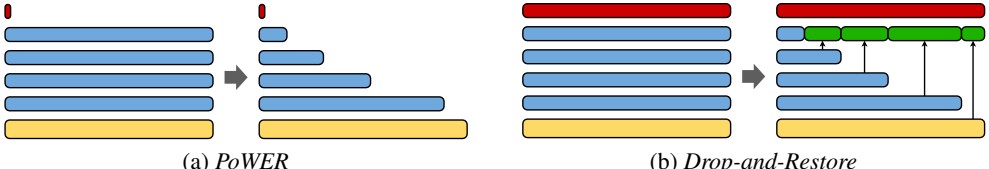

(a) *PoWER*  (b) *Drop-and-Restore*

Figure 1: Illustration of (a) word-vector elimination process in PoWER-BERT (Goyal et al., 2020) and (b) Drop-and-Restore process in Length-Adaptive Transformer. Yellow box and blue boxes imply the output of embedding layer and transformer layers, respectively. Green boxes mean vectors dropped in lower layers and restored at the last layer. Red box is the task-specific layer. Though word-vectors in the middle could be eliminated (or dropped), remaining vectors are left-aligned for the better illustration. In this case, the number of transformer layers is four.

### 3.3 DROP-AND-RESTORE PROCESS

The applicability of the PoWER-BERT, based on which our main contribution above was made, is limited to sequence-level classification, because it eliminates word vectors at each layer. In addition to our main contribution above, we thus propose to extend the PoWER-BERT so that it is applicable to token-level classification, such as span-based question-answering. Our proposal, to which we refer as Drop-and-Restore, does not eliminate word vectors at each layer according to the length configuration but instead sets them aside until the final hidden layer. At the final hidden layer, these word vectors are brought back to form the full hidden sequence, as illustrated graphically in Fig. 1.

### 4 EXPERIMENT SETUP

**Datasets**   We test the proposed approach on both sequence-level and token-level tasks, the latter of which could not have been done with the original PoWER-BERT unless for the proposed Drop-and-Restore. We use MNLI-m and SST-2 from GLUE benchmark (Wang et al., 2018), as was done to test PoWER-BERT earlier, for sequence-level classification. We choose them because consistent accuracy scores from standard training on them due to their sufficiently large training set imply that they are reliable to verify our approach. We use SQuAD 1.1 (Rajpurkar et al., 2016) for token-level classification.

**Evaluation metrics**   We use the number of floating operations (FLOPs) as a main metric to measure the inference efficiency given any length configuration, as it is agnostic to the choice of underlying hardware, unlike other alternatives such as hardware-aware latency (Wang et al., 2020) or energy consumption (Henderson et al., 2020). We later demonstrate that FLOPs and wall-clock time on GPU and CPU correlate well with the proposed approach, which is not necessarily the case for other approaches, such as unstructured weight pruning (Han et al., 2015; See et al., 2016).

**Pretrained transformers**   Since BERT was introduced by Devlin et al. (2018), it has become a standard practice to start from a pretrained (masked) language model and finetune it for each downstream task. We follow the same strategy in this paper and test two pretrained transformer-based language models; $BERT_{BASE}$ (Devlin et al., 2018) and DistilBERT (Sanh et al., 2019), which allows us to demonstrate that the usefulness and applicability of our approach are not tied to any specific architectural choice, such as the number of layers and the maximum length of input sequence. Although we focus on BERT-based masked language models here, the proposed approach is readily applicable to any transformer-based models.

**Learning**   We train a Length-Adaptive Transformer with LengthDrop probability and LayerDrop probability both set to $0.2$. We use $n_s = 2$ randomly sampled intermediate sub-models in addition to the full model and smallest model for applying the sandwich learning rule.

We start finetuning the pretrained transformer without Drop-and-Restore first, just as Goyal et al. (2020) did with PoWER-BERT. We then continue finetuning it for another five epochs *with* Drop-and-Restore. This is unlike the recommended three epochs by Devlin et al. (2018), as learning progresses

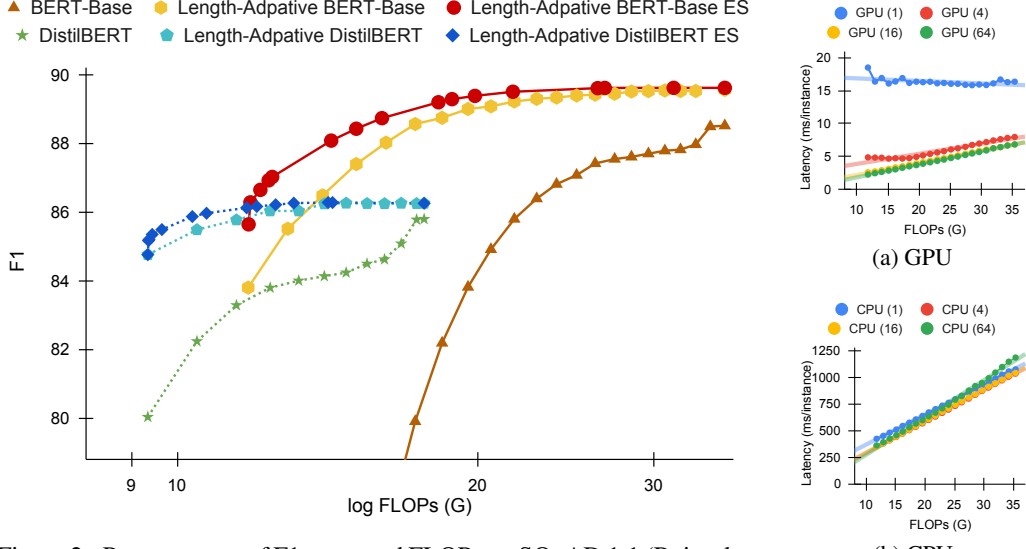

Figure 2: Pareto curves of F1 score and FLOPs on SQuAD 1.1 (Rajpurkar et al., 2016). We apply the proposed method to BERT_Base (solid lines) and DistilBERT (dotted lines). For each model, we draw three curves using (1) standard finetuned transformer with constant-rate length reduction, (2) Length-Adaptive Transformer with constant-rate length reduction, and (3) Length-Adaptive Transformer with length configurations obtained from the evolutionary search.

Figure 3: Correlation between FLOPs and latency with different length configurations.

slower due to a higher level of stochasticity introduced by LengthDrop and LayerDrop. We use the batch size of 32, the learning rate of $5e - 5$ for SQuAD v1.1 and $2e - 5$ for MNLI-m and SST, and the maximum sequence length of 384 for SQuAD v1.1 and 128 for MNLI-m and SST.

**Search**   We run up to $G = 30$ iterations of evolutionary search, using $n_m = 30$ mutated configurations with mutation probability $p_m = 0.5$ and $n_c = 30$ crossover'd configurations, to find the Pareto frontier of accuracy and efficiency.

## 5   RESULTS AND ANALYSIS

**Efficiency-accuracy trade-off**   We use SQuAD 1.1 to examine the effect of the proposed approach on the efficiency-accuracy trade-off. When the underlying classifier was not trained with LengthDrop, as proposed in this paper, the accuracy drops even more dramatically as more word vectors are dropped at each layer. The difference between standard transformer and Length-Adaptive Transformer is stark in Fig. 2. This verifies the importance of training a transformer in a way that makes it malleable for inference-time re-configuration.

When the model was trained with the proposed LengthDrop, we notice the efficacy of the proposed approach of using evolutionary search to find the optimal trade-off between inference efficiency and accuracy. The trade-off curve from the proposed search strategy has a larger area-under-curve (AUC) than when constant-rate length reduction was used to meet a target computational budget. It demonstrates the importance of using both LengthDrop and evolutionary search.

We make a minor observation that the proposed approach ends up with a significantly higher accuracy than DistillBERT when enough computational budget is allowed for inference ($\log$ FLOPs $> 10$). This makes our approach desirable in a wide array of scenarios, as it does not require any additional pretraining stage, as does DistilBERT. With a severe constraint on the computational budget, the proposed approach could be used on DistilBERT to significantly improve the efficiency without compromising the accuracy.

**Maximizing inference efficiency**   We consider all three tasks, SQuAD 1.1, MNLI-m and SST-2, and investigate how much efficiency can be gained by the proposed approach with minimal sacrifice of

| Model | | SQuAD 1.1 | | MNLI-m | | SST-2 | |
|---|---|---|---|---|---|---|---|
| Pretrained Transformer | Method | F1 | FLOPs | Acc | FLOPs | Acc | FLOPs |
| BERT$_{Base}$ | Standard | 88.5 | 1.00x | 84.4 | 1.00x | 92.8 | 1.00x |
| | Length-Adaptive$^\star$ | 89.6 | 0.89x | 85.0 | 0.58x | 93.1 | 0.36x |
| | Length-Adaptive$^\dagger$ | 88.7 | 0.45x | 84.4 | 0.35x | 92.8 | 0.35x |
| DistilBERT | Standard | 85.8 | 1.00x | 80.9 | 1.00x | 90.6 | 1.00x |
| | Length-Adaptive$^\star$ | 86.3 | 0.81x | 81.5 | 0.56x | 92.0 | 0.55x |
| | Length-Adaptive$^\dagger$ | 85.9 | 0.59x | 81.3 | 0.54x | 91.7 | 0.54x |

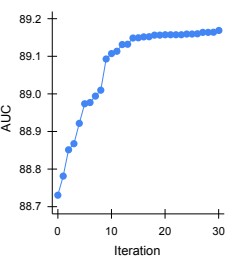

Table 1: Comparison results of standard Transformer and Length-Adaptive Transformer. Among length configurations on the Pareto frontier of Length-Adaptive Transformer, we pick two representative points: Length-Adaptive$^\star$ and Length-Adaptive$^\dagger$ as the most efficient one while having the highest accuracy and the accuracy higher than (or equal to) standard Transformer, respectively.

Figure 4: Example of area under Pareto curve as the evolutionary search of lenth configurations proceeds.

accuracy. First, we look at how much efficiency could be gained without losing on the accuracy. That is, we use the length configuration that maximizes the inference efficiency (i.e., minimize the FLOPs) while ensuring that the accuracy is above or same as the accuracy of the standard approach without any drop of word vectors. The results are presented in the rows marked with Length-Adaptive$^\dagger$ from Table 1. For example, in the case of BERT$_{Base}$, the proposed approach reduce FLOPs by more than half across all three tasks.

From Fig. 2, we have observed that the proposed Length-Adaptive Transformer generalize better than the standard, base model in some cases. We thus try to maximize both the inference efficiency and accuracy, in order to see whether it is possible for the proposed algorithm to find a length configuration that both maximizes inference efficiency and improves accuracy. We present the results in the rows marked with Length-Adaptive$^\star$ from Table 1. For all cases, Length-Adaptive Transformer achieves higher accuracy than a standard transformer does while reducing FLOPs significantly. Although it is not apparent from the table, tor MNLI-m and SST-2, the accuracy of the smallest sub-model is already greater than or equal to that of a standard transformer.

**FLOPs vs. Latency** As has been discussed in recent literature (see, e.g., (Li et al., 2020; Chin et al., 2020)), FLOPs is not a perfect indicator of the real latency measured in wall-clock time, as the latter is affected by the combination of hardware choice and network architecture. To understand the real-world impact of the proposed approach, we study the relationship between FLOPs, obtained by the proposed procedure, and wall-clock time measured on both CPU and GPU by measuring them while varying length configurations. As shown in Fig. 3, FLOPs and latency exhibit near-linear correlation on GPU, when the minibatch size is $\geq 16$, and regardless of the minibatch size, on CPU. In other words, the reduction in FLOPs with the proposed approach directly implies the reduction in wall-clock time.

**Convergence of search** Although the proposed approach is efficient in that it requires only one round of training, it needs a separate search stage for each target budget. It is important for evolutionary search to converge quickly in the number of forward sweeps of a validation set. As exemplified in Fig. 4, evolutionary search converges after about fifteen iterations.

## 6 RELATED WORK

The main purpose of the proposed algorithm is to improve the inference efficiency of a large-scale transformer. This goal has been pursued from various directions, and in this section, we provide a brief overview of these earlier, and some concurrent, attempts in the context of the proposed approach.

**Weight pruning** Weight pruning (Han et al., 2015) focuses on reducing the number of parameters which directly reflects the memory footprint of a model and indirectly correlates with inference speed. However, their actual speed-up in runtime is usually not significant, especially while executing a model with parallel computation using GPU devices (Tang et al., 2018; Li et al., 2020).

**Adaptive architecture**   There are three major axes along which computation can be reduced in a neural network; (1) input size/length, (2) network depth and (3) network width. The proposed approach, based on PoWER-BERT, adaptively reduces the input length as the input sequence is processed by the transformer layers. In our knowledge, Goyal et al. (2020) is the first work in this direction for transformers. More recently, Funnel-Transformer (Dai et al., 2020) and multi-scale transformer language models (Subramanian et al., 2020) also successfully reduce sequence length in the middle and rescale to full length for the final computation. However, their inference complexity is fixed unlike PoWER-BERT because they are not designed for the control of efficiency.

LayerDrop (Fan et al., 2019) drops random layers during the training to be robust to pruning inspired by Huang et al. (2016). Word-level adaptive depth in Elbayad et al. (2019) might seemingly resemble with length reduction, but word vectors reached the maximal layer are used for self-attention computation without updating themselves. Escaping a network early (Teerapittayanon et al., 2016; Huang et al., 2017) based on the confidence of the prediction (Xin et al., 2020; Schwartz et al., 2020; Liu et al., 2020) also offers a control over accuracy-efficiency trade-off, but it is difficult to tune a threshold for a desired computational budget because of the example-wise adaptive computation.

DynaBERT (Hou et al., 2020) can run at adaptive width (the number of attention heads and intermediate hidden dimension) and depth. Hardware-aware Transformers (Wang et al., 2020) construct a design space with arbitrary encoder-decoder attention and heterogeneous layers in terms of different numbers of layers, attention heads, hidden dimension, and embedding dimension.

**Structured dropout**   A major innovation we introduce over the existing PoWER-BERT is the use of stochastic, structured regularization to make a transformer robust to the choice of length configuration in the inference time. Rippel et al. (2014) proposes a nested dropout to learn ordered representations. Similar to LengthDrop, it samples an index form a prior distribution and drops all units having a larger index than sampled one.

**Search**   There have been a series of attempts at finding the optimal network configuration by solving a combinatorial optimization problem. In computer vision, Once-for-All (Cai et al., 2019) use an evolutionary search (Real et al., 2019) to find a better configuration in dimensions of depth, width, kernel size, and resolution given computational budget. Similarly but differently, our evolutionary search is *mutli-objective* to find length configurations on the Pareto accuracy-efficiency frontier to cope with any possible computational budgets. Moreover, we only change the sequence length of hidden vectors instead of architectural model size like dimensions.

## 7   CONCLUSION AND FUTURE WORK

In this work, we propose a new framework for training a transformer once and using it for efficient inference under any computational budget. With the help of training with LengthDrop and Drop-and-Restore process followed by the evolutionary search, our proposed Length-Adaptive Transformer allows any given transformer models to be used with any inference-time computational budget for both sequence-level and token-level classification tasks. Our experiments, on SQuAD 1.1, MNLI-m and SST-2, have revealed that the proposed algorithmic framework significantly pushes a better Pareto frontier on the trade-off between inference efficiency and accuracy. Furthermore, we have observed that the proposed Length-Adaptive Transformer could achieve up to 3x speed-up over the standard transformer without sacrificing accuracy, both in terms of FLOPs and wallclock time.

Although our approach finds *an* optimal length configuration of a trained classifier per computational budget, it leaves open a question whether the proposed approach could be further extended to support per-instance length configuration by for instance training a small, auxiliary neural network for each computational budget. Yet another aspect we have not investigated in this paper is the applicability of the proposed approach to sequence generation, such as machine translation. We leave both of these research directions for the future.

Our approach is effective, as we have shown in this paper, and also quite simple to implement on top of existing language models. We will release our implementation, which is based on HuggingFace's *Transformers* library (Wolf et al., 2019), publicly and plan to adapt it for a broader set of transformer-based models and downstream tasks.

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
