# OpenReview forum: "Length-Adaptive Transformer: Train Once with Length Drop, Use Anytime with Search"
_ICLR.cc/2021/Conference — Reject_

### Official Review · AnonReviewer3 · 2020-10-25
**Simple and intuitive idea, important questions unanswered**

**Rating:** 5
**Confidence:** 4

**Review:**

This paper aims to make inference more efficient for finetuned contextual representation models such as BERT. The authors extend PowerBERT, a method introduced recently to perform  efficient inference by dynamically reducing input tokens as the model goes deeper. The authors address two limitations of PowerBERT: the need to pre-set the required computational budget during training (which makes the model inflexible), and the inability to tackle span level tasks (which require the full sentence at the final layer). The authors present a simple solution to both problems (though one that requires heavy engineering to work, see below), and show promising results compared to several BERT baselines. Overall, the ideas presented in this paper are simple (in a good way), and I would like to see more work that applies similar ideas to gain efficiency and not just accuracy. However, the paper leaves important questions unanswered, and thus I cannot recommend accepting it. First, how expensive is the proposed evolutionary search compared to standard fine-tuning? Second, how does this approach compare to the original PowerBERT model (despite its limitations)? Third, given that the evolutionary search is performed on the validation set, how do the authors evaluate it during model development?
I am curious to read the authors' response, and could be convinced to increase my score, but currently this paper does not meet the ICLR bar.

Detailed summary:

This paper takes PowerBERT, a recently introduced model for efficient inference, and addresses two of its limitations: the need to retrain the model for each computational budget, and the inability to handle span-level tasks such as QA. The authors present simple solutions to both: for the former, they train a model that randomly selects the number of dropped tokens during training, which makes the model more resilient to different levels of pruning, and perform an evolutionary search process to match the exact pruning levels for a given computational budget. For the latter, they restore the tokens dropped at earlier stages in the last layer, allowing the model to access span indices for tokens dropped earlier if necessary. These solutions are solid and seem to work in practice (although they require quite a bit of engineering to work well, see section 3.1). Nonetheless, their implementation leaves a few important questions unanswered:

1. If I understand correctly, the proposed evolutionary search process needs to be run for any computational budget, which makes it suffer from the same problem as the original PowerBERT model. The authors address this concern in section 3.2, saying that "it only require a single pass through the relatively small validation set for each length configuration", but later say they "repeat this iteration G times" (with G=30) and "evolutionary search converges after about fifteen iterations". This leads to the natural question: how much faster is this procedure compared to finetuning the model, as done in PowerBERT? And even if it is faster, it still doesn't result in a completely flexible model, as each new budget level requires rerunning this process.
2. Another issue with this approach, is that the model is (partially) trained on the development set. This leaves me wondering how it was evaluated during model development.
3. The authors build on the PowerBERT model, but never compare to it. Although they (supposedly) provide important benefits compared to this model, it is important to understand at what (accuracy and efficiency) cost.

More comments:
1. This paper would strongly benefit from a proof-read. Several paragraphs and sentences repeat what was just said (e.g., the paragraph starting with "It is not trivial to find an optimal length" on page 2, or the first sentence on page 7). There are multiple typos (e.g., "Lengnth" on Table 1)  and grammatical errors ("it converges rapidly evolutionary search converges after about fifteen iterations.") all around the paper. Moreover, several issues were unclear to me:
a. what are sub-models? are these layers with fewer tokens?
b. how is model distillation incorporated in the solution (last paragraph of section 3.1)?
2. BERT uses word-pieces and not words.

---

> ### Author Response · Authors · 2020-11-25
> **Response to Reviewer3**
>
> Q1. Clarification about evolutionary search and its cost compared to standard fine-tuning
>
> A1. We revised the explanation about the proposed evolutionary search (the first sentence of Section 3.2) to make the reviewer’s understanding clear. We do not rerun the evolutionary search process multiple times for different computational budgets. Evolutionary search is only performed once after training a Length-Adaptive Transformer with LengthDrop. This search process results in numerous length configurations. We can choose one of those configurations depending on the computational budget given at test time. In sum, our model is completely flexible as each new budget level does not require rerunning the evolutionary search.
>
> In our expression, "it only requires a single pass through the relatively small validation set for each length configuration" means that the evaluation for each length configuration requires inference on the validation set rather than additional training like other NAS methods.
> "repeat this iteration G times" (with G=30) and "evolutionary search converges after about fifteen iterations" mean that the evolutionary search consists of multiple iterations (in other words, generations doing mutations, cross-overs, and selection of configurations on the Pareto curve)
>
> The evolutionary search is a separate process (not training) in addition to the finetuning, and it is quite fast because it only requires evaluation on the validation set. This evaluation is repeated for multiple length configurations considered during the search. The number of evaluated length configurations is G * (n_m + n_c). For instance, the evolutionary search on sentence classification tasks takes only a few hours. Moreover, the evolutionary search can be improved much faster since most of the steps are parallelizable, though we have not implemented it yet.
>
> --------------------------------------------------
>
> Q2. Evaluation method of evolutionary search
>
> A2. Like the previous question, we want to clarify that the evolutionary search does not belong to the training. The evolutionary search can be regarded as a model selection because a length configuration can be regarded as a model hyperparameter. Therefore, we used the development set to search for length configurations.
>
> --------------------------------------------------
>
> Q3. Comparison with the original PoWER-BERT
>
> A3. Although we did not directly compare to PoWER-BERT in terms of accuracy-efficiency trade-off, we believe that our method is superior to PoWER-BERT because of the regularization effect from training with LengthDrop that takes into account multiple length configurations jointly. We will add comparison experiments with PoWER-BERT to the final version.
>
> For indirect comparison, according to the PoWER-BERT paper, PoWER-BERT achieves 2.6x speedup for MNLI-m and 2.4x speedup for SST-2 by losing 1% of their accuracy. On the other hand, we obtained a 2.9x speedup without losing accuracy on MNLI-m and SST-2. Of course, we know that their speedup is based on inference time and ours is based on FLOPs, so it may not be a fair comparison. However, their time measurement was done with a batch size of 128 on GPU, so our speedup in execution time would be close to 2.9x, considering Figure 3.
>
> Moreover, our Length-Adaptive Transformer provides far more important benefits (anytime prediction) than PoWER-BERT, as you mentioned.
>
> --------------------------------------------------
>
> Q4. Clarification about sub-models, distillation, and word
>
> A4. We added the clarification about sub-models and distillation in the revision (last paragraph of Section 3.1). Sub-models mean models with the length reduction, and they share the same model parameters with the full model without length reduction.
>
> Distillation in the last paragraph of Section 3.1 means inplace distillation that makes sub-models’ prediction close to the full model’s prediction.
>
> Yes, we notice that BERT uses word-pieces (subwords) as a basic token. We use the term “word” as a basic token in the vocabulary to represent a sentence like PoWER-BERT paper and many other NLP papers.
>
> --------------------------------------------------
>
> Q5. Whether the proposed solutions require heavy engineering
>
> A5. We disagree that our method requires heavy engineering to work. Our methods, especially Section 3.1, are quite simple and easy to implement. They can be integrated into any training pipeline or model architecture with few lines of additional code.

---

### Official Review · AnonReviewer1 · 2020-10-28
**Lacks experiments**

**Rating:** 5
**Confidence:** 4

**Review:**

##########################################################################

Summary:

The paper proposed the Length-Adaptive Transformer. The model can be trained once and directly applied to different inference scenarios. To achieve this goal, the author proposed the LengthDrop method, which randomly samples the length at each layer. In addition, the author used the sandwich rule to train the model. At each step, the sandwich rule will train the largest model, the smallest model, and another bunch of randomly sampled models. In the inference phase, the paper proposed to search for the best length configuration that balances the accuracy and latency tradeoff via evolutionary search. Moreover, to generalize the model to token annotation tasks, the author proposed the Drop-and-Restore process, in which the tokens that have been dropped are used again in the final layer. Experiments show that Length-Adaptive Transformer is able to outperfom the baseline models when evaluated at the same latency level.


##########################################################################

Reasons for score:


The paper is well-written and the length-adaptive idea is reasonable. However, I think it still requires more experiments. Also, the training process is not very clear to the reader. It is not so clear how different techniques impact the final performance and the author has not reported the training time. Due to these two reasons, I choose to vote for weak rejection.


##########################################################################

Pros:


1. The idea of length-adaptive transformer is novel. Perform on-demand truncation of the hidden lengths has not been well explored.


##########################################################################

Cons:


1. The paper needs to conduct experiments on more text classification dataset. Currently, only SST-2 and MNLI are considered. Also, the paper lacks the comparison with "[NeurIPS2020] DynaBERT: Dynamic BERT with Adaptive Widthand Depth", which is also able to balance the latency and accuracy.

2. The training process is quite complicated and involves multiple steps, e.g., Length-Drop, LayerDrop, and Sandwich rule. The author has not explained the relative contribution of each techniques. Also, the author has not reported the training time of the whole pipeline.


##########################################################################

Questions during rebuttal period:


Please address and clarify the cons above


#########################################################################

Typos:

(1) Section 2.2, Paragraph 2, "more efficient and more accuracy" should be "more efficient and more accurate"
(2) Page 7, under Table 1 and Figure 4, "Lengnth" should be "Length"

---

> ### Author Response · Authors · 2020-11-25
> **Response to Reviewer1**
>
> Q1. Experiments on text classification dataset more than MNLI and SST-2
>
> A1. Experiments on three datasets (SQuAD v1.1, MNLI, and SST-2) cover different types of NLP tasks, including question answering, textual entailment (sentence pair classification), and sentiment classification (single sentence classification). Thus, we think that it would be enough to verify that our method works well. However, we will add experimental results on other datasets in GLUE benchmark to the final version.
>
> --------------------------------------------------
>
> Q2. Comparison with DynaBERT
>
> A2. We compared DynaBERT and other adaptive computation works in our “Related Work” section. We want to emphasize that those methods are orthogonal with ours, meaning that various adaptive dimensions (sequence length, depth, attention head, hidden dimension, etc.) can be jointly used. In other words, even if other adaptive methods show better curves than ours, our method and theirs can boost each other when combined. Our adaptive sequence length idea is novel and empirically proven effective by itself.
>
> Anyway, we are happy to compare and combine our method with DynaBERT as future work. However, we want to note that DynaBERT has been published in NeurIPS 2020 very recently and released their implementation to reproduce their results two months ago.
>
> According to the results of the DynaBERT paper, we can make an indirect comparison. It turned out that we achieved comparable (or better in some cases) performance with them. On SQuAD v1.1, both ours and DynaBERT can reduce FLOPs to about half of the standard model without losing accuracy. Followings are how much each method improves the best accuracy from the standard method on multiple datasets:
>
> SQuAD v1.1 F1 - Ours: 88.5 -> 89.6 (+1.1) / DyanBERT: 88.7 -> 89.7 (+1.0)
>
> MNLI-m accuracy - Ours: 84.4 -> 85.0 (+0.6) / DynaBERT: 84.8 -> 84.9 (0.1)
>
> SST-2 accuracy - Ours: 92.8 -> 93.1 (+0.3) / DynaBERT: 92.9 -> 93.3 (+0.4)
>
> --------------------------------------------------
>
> Q3. Complicated training process and their relative contribution
>
> A3. Although our training process requires several techniques, they are easy to implement and integrated into any training pipeline or model architecture with few lines of additional code. In our preliminary experiments, we observed that removing LayerDrop or sandwich rule drops the performance. We will add an ablation study to the final version.
>
> --------------------------------------------------
>
> Q4. Report of the training time
>
> A4. We agree that efficient training is important. Nonetheless, it is worth taking more time (only a few more hours with a single GPU) considering trained models can be used multiple times after a single period of training, even if the training method can increase the accuracy a little. For example, RoBERTa is finetuned with 10 epochs to find the best model, although it is slightly better than finetuning with 3 epochs as BERT did.
>
> Our method's training time is determined by many factors, including LengthDrop probability, LayerDrop probability, the number of sandwiches, and the number of training epochs. As you can imagine, higher drop probability will make computation faster, but the number of sandwiches decides the number of sub-models that require additional forward steps. We used both LengthDrop probability and LayerDrop probability to 0.2 and the number of sandwiches to 2. We trained with more epochs due to the regularization effect of LengthDrop and LayerDrop. Our training method with 5 epochs took 2-4 times longer than standard training with 3 epochs in our experiments.
>
> However, compared to PoWER-BERT, which requires the length configuration search stage and separate finetuning steps per different length configurations, our method is much more efficient in terms of training time.

---

### Official Review · AnonReviewer2 · 2020-10-28
**Official Blind Review #2**

**Rating:** 4
**Confidence:** 4

**Review:**

This work introduces a method, called LengthDrop, to train a Length-Adaptive Transformer that supports adaptive model architecture based on different latency constraints. In order to make the model robust to variable input lengths, the method stochastically reduces the length of a sequence at each layer during training. Once the model is trained, the method uses an evolutionary search to find subnetworks that maximize model accuracy under a latency budget.

Pros:
- Accelerating the inference speed of Transformer networks is an important problem.
- The idea of training a length-adaptive Transformer once and using it in different scenarios with different latency constraints is interesting.

Cons:
- The discussion with several state-of-the-art work is lacking.
- The experimental setup is vague, and the evaluation results are inadequate.

The paper looks from an interesting angle to build adaptive Transformers for inference -- reducing the input sequence at each Transformer layer. However, there are a few concerns.

First, the paper proposes to use a series of techniques to make LengthDrop work but lacks the ablation studies to show how those techniques help to make Transformer length adaptive. For example, Section 3.1 states that LengthDrop requires LayerDrop[1], which also supports adaptive Transformer by stochastically dropping layers during training for adaptive inference. However, there are no ablation studies or comparison results on LengthDrop vs. LayerDrop in terms of the accuracy-vs-latency trade-off. This raises the question of whether LengthDrop is necessary to obtain the given accuracy-vs-latency or perhaps simpler alternatives such as LayerDrop would be sufficient.

In addition to LayerDrop, it appears that the paper also incorporates several other fixes, such as the sandwidth rule and inplace distillation, which are borrowed from prior work.  However, how these fixes contribute to LengthDrop is not clearly explained, and there are no studies nor experimental results to explain how each technique contributes to the final accuracy-vs-latency results.

Second, the comparison with related work is weak. In particular, LengthDrop is built on top of PoWER-BERT, yet the evaluation does not compare with PoWER-BERT.  Furthermore, the paper compares with DistillBERT, but there are multiple knowledge distillation based work that show better performance than DistillBERT, such as TinyBERT[2]. In terms of adaptive architecture, the evolutionary search of length configurations is similar to the NAS process in the Hardware-aware Transformer[2], which seems to be very related as it also uses evolutionary search to find a specialized sub-network of Transformer models with a latency constraint. The paper briefly mentioned [2], but it is not clear the advantage of this work as compared with [2].

Third, the paper lacks enough information on the evaluation setups, raising several questions on the reported speedups. For example, it is unclear what's the batch size used in the evaluation. Figure 3(a) shows that reducing FLOPs on GPU does not lead to a reduction of latency for batch size 1, which is the common setting for online inference scenarios as queries come in one-by-one. It is unclear whether input length reduction may actually bring significant latency reduction when the batch size is small (e.g., 1), as the large matrix multiplications have been highly optimized on modern CPU and GPU through efficient kernels (e.g., cuDNN). Even for results on CPU and GPU with batch size >= 16, it is less clear whether the linear correlation between FLOPs and latency is a fact of failing to use highly optimized BLAS libraries, because the paper does not report the details on the hardware, the inference frameworks, and libraries it uses for the experimental results.

In addition to the batch size and lack of hardware/platform/library information, the experimental setup for training a Length-Adaptive Transformer is also not very clear. For example, it is unclear what's the maximum sequence length is used in training. Whether mixed sequence length is used in training BERT?  What is the sequence length(s) to obtain the results in Table 1? Without associating the actual length reduction ratio, it is hard to evaluate the reported FLOPs reduction.

[1] Fan et. al. "Reducing Transformer Depth on Demand with Structured Dropout", https://arxiv.org/abs/1909.11556

[2] Jiao et. al. "TinyBERT: Distilling BERT for Natural Language Understanding", https://arxiv.org/abs/1909.10351

---

> ### Author Response · Authors · 2020-11-25
> **Response to Reviewer2**
>
> Q1. Comparison with related work
>
> A1. We argue that other adaptive computation methods (like DynaBERT and Hardware-aware Transformer) and distillation methods (like TinyBERT) are orthogonal with our adaptive sequence length method. Please check indirect comparison in our response with DynaBERT (Reviewer1 - Answer2) and PoWER-BERT (Reviewer3 - Answer3). We will add more comparison results to the final version.
>
> --------------------------------------------------
>
> Q2. Ablation studies of training techniques
>
> A2. LayerDrop alone during the finetuning does not help, similar to FAQ #3 in the official LayerDrop repository (https://github.com/pytorch/fairseq/tree/master/examples/layerdrop#faq). All of LayerDrop, sandwich rule, and inplace distillation were significantly helpful when used during the training with LengthDrop. We will add an ablation study by removing each technique one-by-one to the final version.
>
> --------------------------------------------------
>
> Q3. Evaluation/experimental setups
>
> A3. We evaluate speedup in terms of FLOPs reduction instead of measuring latency in a specific setting because, as you mentioned in the review, the speedup is heavily determined by many other factors, including batch size and hardware/platform/library. We also notice that real execution time is important. Therefore, we measured a correlation between FLOPs and latency in our setting as an example (Figure 3). Our implementation is based on the PyTorch version of HuggingFace Transformers library. For Figure 3, we used Tesla V100 GPUs and Xeon CPUs. We believe that future low-level engineering could lead to further gain in a favorable way to sequence length reduction.
>
> We added finetuning hyperparameters (batch size, learning rate, and maximum sequence length) used for experiments in the revision. We use the batch size of 32, the learning rate of 5e-5 for SQuAD v1.1 and 2e-5 for MNLI-m and SST, and the maximum sequence length of 384 for SQuAD v1.1 and 128 for MNLI-m and SST, as usually used for BERT finetuning.

---

### Official Review · AnonReviewer4 · 2020-10-28
**Interesting direction, reasonable and interesting techniques, limited experiments and unclear source of gain**

**Rating:** 6
**Confidence:** 5

**Review:**

The work targets an interesting direction of improving the efficiency of Transformers by reducing the sequence length. The main contributions of the work are (1) proposing LengthDrop as the way to achieve length reduction; (2) utilizing techniques developed in NAS, namely one-shot NAS, to enable proper training and allow adaptive drop ratio search after training. All these ideas are very reasonable and interesting. Empirically, the authors show that the proposed method is able to match or even outperform BERT-base model with 1/3 - 1/2 FLOPs during inference (not training).

The first concern I have is that the source of the gain is unclear in two aspects:
(1) The proposed model is finetuned for longer (5 + 5 epochs) compared to the 3 epochs of the original BERT. However, the baseline numbers are still based on 3 epochs. This could be one of the reasons why the proposed model even outperform the original BERT.
(2) With the "Inplace Distillation" in play, the obtained model is effectively a distilled model. This adds another layer of complication to judge how much the gain/loss comes from distillation and the length reduction.

Secondly, authors do not mention much about the training (finetuning + ES) cost compared to the standard BERT or Power-BERT. In many real-world cases, this cost is also non-trivial. This may be part of the reason why only 3 datasets are considered in this paper.

---

> ### Author Response · Authors · 2020-11-25
> **Response to Reviewer4**
>
> Q1. Finetuning longer might be the reason for better accuracy
>
> A1. In our preliminary experiments for the fair comparison, increasing the number of training epochs longer than three does not improve (or sometimes worsen) the accuracy of the original BERT. We presume that this is the reason why the original BERT paper finetuned BERT with only three epochs. We finetune longer to train our model because of the regularization effect of LengthDrop. It is generally known that more regularization usually requires more training steps, as we try to explain in the “Learning” part of Section 4.
> Moreover, we want to emphasize that we aim to propose a novel method that allows anytime prediction with adaptive sequence length rather than higher accuracy, although our method even outperforms the standard training baseline.
>
> --------------------------------------------------
>
> Q2. How much the gain/loss comes from distillation and the length reduction
>
> A2. We want to clarify that “inplace distillation” does not require a separate distilled model. That is what *inplace* means. Inplace distillation is different from other standard knowledge distillation that produces another (usually smaller) model.
>
> A sub-model implies a model with length reduction, and it shares the same model parameters with the full model without length reduction. Inplace distillation makes sub-models’ prediction close to the full model’s prediction. It is used to train sub-models with length reduction effectively by giving a richer training signal rather than a supervised loss from a one-hot label. Inplace distillation was helpful to make the model robust to the length reduction.
>
> --------------------------------------------------
>
> Q3. Training cost compared to the standard BERT or Power-BERT
>
> A3. Please check our discussion in our response about training time (Reviewer1 - Answer4  & Reviewer3 - Answer1) and the used datasets (Reviewer1 - Answer1).

---

### Public Comment · ~Deming_Ye1 · 2020-11-11
**Delete padding?**

In MNLI, the average context length is about 29. In your experiment, are the major pruning words the padding?

---

### Author Response · Authors · 2020-11-25
**General Response**

We are glad to see that most reviewers feel our method is simple, intuitive, novel, interesting, and practically useful. We revised our manuscript by fixing typos and adding some clarifications to help better understanding. Concerns from the reviewers are addressed in our response individually. We are planning to add more extensive experimental results to the camera-ready version.

We argue that the lack of (1) comparison with other SOTA methods according to the accuracies alone or (2) ablation study of our main components does not invalidate the efficacy of the proposed approach. First, our goal has been to propose a new anytime prediction method with adaptive sequence length, that can be used with any transformer-based classifier rather than only with a particular classifier that is state-of-the-art in terms of the classification accuracy, although we were able to demonstrate the proposed approach still attains a good accuracy level. Second, we found that training techniques are significantly crucial to the final performance in our preliminary experiments and believe that combining them to our method works is one of our contributions.

We strongly believe that our work contributes meaningfully to the machine learning and natural language processing community as a promising research direction and also has a potential for impact in the industry.

---

### Decision · Program_Chairs · 2021-01-07
**Final Decision**

**Decision:**

Reject

**Comment:**

The paper attempts to reduce computational cost of Transformer models. In this regard, authors generalizer PoWER-BERT by proposing a variant of dropout that reduces training cost by randomly sampling a fraction of the length of a sequence to use at each layer. Further, a sandwich training method is used which trains a spectrum of randomly sampled model between the largest and the smallest size model. At test time, the best length configuration that balances the accuracy and latency tradeoff via evolutionary search is used. The reviewers found the general idea interesting, but raised a number of concerns. First, proper baselines should be used and related works be discussed. In particular, the method is built on top of Power-BERT, yet it does not directly compare with it, and there was no good response when pointed out by a reviewer. Second, as the paper employs many tricks (some new some from prior work), but does not do any ablation studies to show how each of those contributes to the final accuracy gains. Finally, to showcase benefit compared to prior works in terms of computational cost a proper evaluation methodology and actual speedups for batch size 1 inference should be provided. Thus, an improved evaluation would benefit the paper a lot and paper in its current form is not ready for publication.